# Tuberculosis Treatment Response Monitoring by the Phenotypic Characterization of *MTB*-Specific *CD4+* T-Cells in Relation to HIV Infection Status

**DOI:** 10.3390/pathogens11091034

**Published:** 2022-09-12

**Authors:** Nádia Sitoe, Mohamed I. M. Ahmed, Maria Enosse, Abhishek Bakuli, Raquel Matavele Chissumba, Kathrin Held, Michael Hoelscher, Pedroso Nhassengo, Celso Khosa, Andrea Rachow, Christof Geldmacher

**Affiliations:** 1Instituto Nacional de Saúde, Marracuene 3943, Mozambique; 2CIH LMU Center for International Health, Ludwig-Maximilians University, 80802 Munich, Germany; 3Division of Infectious Diseases and Tropical Medicine, Klinikum of the University of Munich (LMU), 80802 Munich, Germany; 4German Center for Infection Research, Partner Site Munich, 80802 Munich, Germany

**Keywords:** tuberculosis, HIV, *Mycobacterium tuberculosis* (*MTB*)-specific, lung severity

## Abstract

HIV infection causes systemic immune activation, impacts TB disease progression and hence may influence the diagnostic usability of *Mycobacterium tuberculosis*-specific T cell profiling. We investigated changes of activation and maturation markers on *MTB*-specific CD4+ T-cells after anti-tuberculosis treatment initiation in relation to HIV status and the severity of lung impairment. Thawed peripheral blood mononuclear cells from TB patients with (*n* = 27) and without HIV (*n* = 17) were analyzed using an intracellular IFN-γ assay and flow cytometry 2 and 6 months post-TB treatment initiation. H37Rv antigen was superior to the profile *MTB*-specific CD4+ T-cells phenotype when compared to PPD and ESAT6/CFP10. Regardless of HIV status and the severity of lung impairment, activation markers (CD38, HLA-DR and Ki67) on *MTB*-specific CD4+ T-cells declined after TB treatment initiation (*p* < 0.01), but the expression of the maturation marker CD27 did not change over the course of TB treatment. The *MTB*-specific T cell phenotype before, during and after treatment completion was similar between people living with and without HIV, as well as between subjects with severe and mild lung impairment. These data suggest that the assessment of activation and maturation markers on *MTB*-specific CD4+ T-cells can be useful for TB treatment monitoring, regardless of HIV status and the severity of lung disease.

## 1. Introduction

Tuberculosis (TB) is a disease caused by *Mycobacterium tuberculosis* bacilli (*MTB*) [1,2]. In 2020, 10 million people were diagnosed with TB globally, resulting in 1.5 million deaths, 214,000 of which were recorded among people living with HIV. TB is an opportunistic infection in HIV patients, particularly in the countries of sub-Saharan Africa [1]. Mozambique is among the top 30 countries globally with the highest burden of TB disease [1]. In 2020, the TB incidence in the Mozambican population was 368 per 100,000, with HIV coinfection accounting for 27% of TB cases [1]. Particularly in the absence of antiretroviral therapy (ART), HIV-1 infection represents a major risk factor, increasing the risk to develop active TB (ATB) by 26-fold [3]. Studies have shown the importance of *MTB*-specific CD4+ T helper-1 cell responses in the control of *MTB* infection [4]; however, these cells are rapidly depleted from circulation during HIV infection [5,6]. Furthermore, the overall depletion of CD4+ T-cells caused by HIV-1 infection correlates with the risk of developing ATB infection [3,4,7,8]. ART initiation partially restores *MTB*-specific CD4+ T-cell responses and simultaneously reduces the risk of TB disease progression in people living with HIV [9].

The utility of activation and maturation markers expressed by *MTB*-specific CD4+ T-cells for TB diagnosis and treatment monitoring in peripheral blood has been well described [10,11,12,13,14]. Assessing the expression of the activation and proliferation markers CD38, HLA-DR and Ki-67 can assist in discriminating between ATB and latent TB infection (LTBI) and can be used for monitoring TB treatment response [10,11,12,13,14,15] and, potentially, for the early detection of TB disease progression [16]. TB treatment also induces the reduction of activation markers on *MTB*-specific CD4+ T-cells in HIV/TB patients, indicating that these markers could serve as a TB diagnosis tool regardless of HIV status [17,18,19]. Together, these studies suggest that HIV infection does not per se have a major impact on the phenotypic profiles of *MTB*-specific T cells utilized for the diagnostic discrimination of ATB versus LTBI or cured TB.

The lungs are among the major organs affected by TB disease; more than 40% of treated pulmonary TB patients develop chronic lung impairment [20,21], as shown by chest radiography (CXR) abnormalities [22] and the reduced capacity in lung functional tests or physical exercises [23]. The risk factors associated with pulmonary function deterioration include smear-positive disease, extensive pulmonary involvement prior to anti-tuberculosis treatment, prolonged anti-tuberculosis treatment and limited radiographic improvement after treatment [24]. Apart from this, *MTB*-specific T cell functional profiles have been linked to pulmonary cavities in several studies [13,25,26]. TNF-α and IFN-γ producing *MTB*-specific T cells within both the CD4+ and CD8+ T-cell compartments were significantly reduced in TB patients with cavities as compared to those with mere lung infiltrates [26]. Furthermore, Fan et al. (2015) has shown that *M. tuberculosis* antigen-specific Th1 response decreases when pulmonary TB lesions develop to severe cavities [25].

Systemic immune activation—assessed on circulating CD8+ or CD4+ T-cells by the co-expression of HLA-DR and CD38—is a hallmark of HIV [26,27,28,29] and TB [13,19] infections and correlates with more rapid HIV disease progression [27]. Indeed, systemic immune activation correlates with CD4+ T-cells depletion [27], is a major determinant of survival in advanced HIV-1 disease [29], and may have utility in the clinical management of HIV-infected persons [30]. To what degree HIV influences the activation profile of *MTB*-specific CD4+ T-cells as bystander activation or because of the higher pathogen activity associated with HIV infection at, before or after ending TB treatment is currently unclear. This is of particular interest for the expression of the activation marker CD38, which differentiated best in previous studies between active TB and LTBI in HIV negative TB patients [11,12]. Furthermore, there are gaps concerning the diagnostic usefulness of *MTB*-specific T cell phenotypic assessments in HIV/TB coinfection in relation to lung damage and the severity of lung impairment before and after TB treatment. Based upon observations that treatment-induced reductions in *MTB*-specific T cell activation correlate with the time to sputum-culture conversion in HIV-negative TB patients [11], we hypothesized that reduction in the activation marker expression of *MTB*-specific CD4+ T-cells is primarily driven by TB treatment, reflecting reductions in the in vivo bacterial load regardless of HIV status and the severity of lung impairment. 

Here, we therefore studied the dynamics of *MTB*-specific CD4+ T-cells’ phenotypic characteristics in relation to TB treatment, the severity of lung impairment and clinical outcomes in well characterized TB patients in a Mozambican cohort in the context of HIV coinfection and lung function impairment. 

## 2. Results

### 2.1. Characteristics of the Study Participants

We had valid results from 44 participants and stratified these based on HIV status into TB monoinfected (*n* = 17) and HIV/TB coinfected (*n* = 27) groups. The median age was 37 years [IQR: 30.6–48.1] and 39 years [IQR: 27.4–43.9] for the HIV/TB coinfected and TB monoinfected groups, respectively (Table 1). Most of the TB patients were male (31 out of 44) and had converted to TB culture negativity at month 2 (M2) (38/44), reflecting the pattern also observed in the main TB sequel cohort. The remaining TB culture-positive patients were negative by the end of TB treatment (month 6). The median ratio monocyte/lymphocyte was similar among the HIV/TB coinfected and TB monoinfected groups—0.47 [IQR: 0.37–0.88] and 0.41 [IQR: 0.38–0.63], *p* = 0.99, respectively. The median levels of AST and ALT were also similar between the two groups, *p* = 0.9041 and *p* = 0.6041, respectively. The median Ralph score (RS) at baseline (BL) was 15.0 [IQR: 10.0–45.0] for the HIV/TB coinfected group and for the TB monoinfected group [IQR: 10.0–35.0]. At month 6 (M6), the median Ralph score was higher in the TB monoinfected group (RS = 10.0) compared to the HIV/TB coinfected group (RS = 6.0), but the difference was not statistically significant (*p* = 0.1). Twenty-nine of the TB patients analyzed had lung function impairment at baseline according to spirometry, and the impairment was not completely resolved at the end of TB treatment (Table 1).

### 2.2. TB Treatment Reduces the Expression of Activation Markers on MTB-Specific CD4+ T-Cells but Not on Total CD4+ T-Cells, Regardless of HIV Status

As described above, we considered samples that had at least 20 CD4 IFN-γ + T-cell counts for the analyses of phenotypic characteristics. Overall, the TB monoinfected subjects showed a higher probability of responding to *MTB*-specific peptides as compared to the HIV/TB coinfected subjects. The average frequencies of responders with valid results after stimulation with H37Rv, Purified Protein Derivative (PPD) and Early Secretory Antigen Target Protein 6/Culture Filtrate Protein 10 (ESAT-6/CFP-10) were 92.2%, 82.4% and 49.0% for the TB monoinfected group and 78.8%, 62.9% and 55.6% for the HIV/TB coinfected group, respectively (Appendix A). Hence, the stimulation with antigen H37Rv resulted in the most robust detection of *MTB*-specific T cell responses, and we therefore focused our analyses of *MTB*-specific T-cell phenotypes on this antigen.

We then compared the frequencies of activation (CD38, HLA-DR and Ki67) and maturation (CD27) markers on *MTB*-specific CD4+ T-cells (representative dot plots presented in Appendix A) at baseline, month 2 and month 6 between the HIV/TB coinfected and TB monoinfected subjects and within each group. Overall, combining both the TB monoinfected and HIV/TB coinfected groups, we observed that the expression of each of the activation markers was significantly reduced as early as 2 months into treatment, while the expression of the maturation marker CD27^pos^ did not change much over time (Figure 1A–D). We also compared the HIV/TB coinfected and TB monoinfected groups at each timepoint (Figure 1E–H). At baseline, we observed that the frequencies of *MTB*-specific CD4+ T-cells expressing CD38 (median: 37.80% versus 27.0%, *p* = 0.4), HLA-DR (median: 17.40% versus 18.70, *p* = 0.9) and Ki67 (median: 11.84% versus 6.5%, *p* = 0.1) were largely comparable between HIV/TB coinfected and TB monoinfected patients. Additionally, at the same time point, the frequencies of *MTB*-specific T cells with a CD27^pos^ phenotype in the HIV/TB coinfected group (median: 14.25%) were slightly lower than those in the TB monoinfected group (median: 22.90%); however, the difference was not statistically significant (*p* = 0.08).

The frequencies of the *MTB*-specific CD4+ T-cells expressing CD38 were significantly reduced from baseline (median: 37.8%) to month 2 (median: 15.20%) (*p* = 0.0079) among HIV/TB coinfected patients, while reductions in the TB monoinfected group were relatively minor but statistically significant (median: 26.8% to 18.5%, *p* = 0.0052). At month 6, significant reductions were observed in both the HIV/TB coinfected (median: 37.80% to 7.45%, *p* = 0.0171) and TB monoinfected (median: 26.80% to 3.60%, *p* = 0.001) groups compared to baseline. These changes in the frequencies of *MTB*-specific CD4+ T-cells were not observed in the total CD4+ T-cells expressing CD38 or HLA-DR. The profile was similar over 6 months of TB treatment in both the HIV/TB coinfected and TB monoinfected subjects (Appendix A). We also observed that five of the six TB patients who were still sputum-culture *MTB*-positive at month 2 had substantially decreased frequencies of *MTB*-specific CD4+ T-cells that expressed CD38.

Additionally, we correlated the proportion of the *MTB*-specific CD4+ T-cells expressing the three activation markers, CD38, HLA-DR and Ki67, and the maturation marker, CD27, confirming the previous results reported by Ahmed and colleagues that a positive correlation between the activation markers is present [11]. The strongest correlation was observed between CD38 and HLA-DR expression on *MTB*-specific CD4+ T-cells (r = 0.78 and *p* < 0.0001 in TB monoinfection) (Appendix A). HIV infection slightly decreased the strength of correlation of HLA-DR^pos^ with CD38^pos^ and Ki67^pos^ T cells (r = 0.34, *p* = 0.04 and r = 0.33, *p* = 0.0427, respectively) and increased the strength of correlation of CD38^pos^ with Ki67^pos^ T cells (from r = 0.49 to r = 0.66). No correlation was observed between CD38 and CD27 expression on *MTB*-specific CD4+ T-cells, regardless of HIV status (r = −0.0002, *p* = 0.99 for TB monoinfected and r = −0.26, *p* = 0.12 for HIV/TB coinfected). Together, these results mostly confirm our previous results in HIV negative TB patients and showed that active TB patients, in our setting, typically have activated *MTB*-specific T cells, which decrease rapidly after TB treatment initiation, regardless of HIV infection.

We then intended to address whether the level of reduction in T-cell activation between baseline and month 2 was specific to the *MTB*-specific CD4+ T-cell compartment in both HIV-positive and HIV-negative TB patients. Overall, there were no differences observed in CD38, HLA-DR and Ki67 expression within the total CD4+ T-cells between baseline and month 2 in either group (Figure 2). The expression dynamics for these three markers were significantly different between the total and *MTB*-specific CD4+ T-cells (CD38^pos^, Ki67^pos^ and HLA-DR^pos^, *p* = 0.003, *p* = 0.02 and *p* = 0.03, respectively) for the HIV/TB coinfected. Only the slope of the *MTB*-specific T cells expressing HLA-DR was significantly lower than the total CD4+ T-cells in the TB monoinfected subjects (*p* = 0.02) (Figure 2). Hence, the reduction in cellular activation between baseline and month 2 upon TB treatment initiation was specific for the *MTB*-specific T cell compartment and was more pronounced in the HIV/TB coinfected patients.

### 2.3. Frequency of MTB-Specific CD4+ T-Cells Expressing the Activation Markers Reduces over 6 Months of TB Treatment Regardless of the Severity of Lung Impairment

Lung impairment can be a consequence of pulmonary TB infection. Using spirometry, we measured the lung impairment at the end of treatment (month 6). We categorized subjects according to their end-of-treatment lung impairment into “severe” (*n* = 13), “moderate” (*n* = 4) and “mild” (*n* = 11) and compared the phenotypic CD4+ T-cells profile at each time point (baseline, month 2 and month 6).

Overall, we observed that the expression of the activation markers CD38 and HLA-DR on *MTB*-specific CD4+ T-cells was significantly reduced after 6 months of TB treatment (*p* = 0.003 for CD38, *p* = 0.0027 for HLA-DR and *p* = 0.0020 for Ki67), and all these markers were already significantly reduced at month 2. The expression of the maturation marker CD27 on *MTB*-specific CD4+ T-cells did not change much over the observed time (*p* = 0.31).

Subjects with severe lung impairment after treatment completion had slightly higher frequencies of MTB-specific CD4+ T-cells expressing CD38, HLA-DR and Ki67 compared to subjects with mild impairment, but these differences were not statistically significant (Figure 3A–C). Additionally, for the expression of activation and maturation markers among the severe or mild lung impairment groups, we observed that only the frequency of MTB-specific CD4+ T-cells expressing HLA-DR among subjects with mild lung impairment did not significantly change over 6 months of TB treatment (median: 13.85% [IQR: 3.08–28.53%] at baseline to 4.43% [IQR: 1.25%–7.72%] at month 6, *p* = 0.10). The median frequencies of *MTB*-specific CD4+ T-cells expressing CD27 did not alter over the duration of TB treatment for subjects with severe (*p* = 0.92) and mild lung impairment (*p* = 0.054) (Figure 3D). 

## 3. Discussion

Indeed, some data indicate that TB patients living with HIV may be affected by more continuous *MTB*-specific T cell activation, even at the end of treatment, and hence have a disease that is probably not completely resolved. Our present study addresses the dynamics of the expression of activation and maturation markers on *MTB*-specific CD4+ T-cells in well characterized pulmonary TB patients upon TB treatment initiation in a high TB and HIV endemic setting. We focused on the potential impact of HIV coinfection and correlations with lung function impairment.

Our study tested PPD, an ESAT-6/CFP-10 peptide pool and the H37Rv antigen; the H37Rv antigen stimulation enabled the detection of a maximum of IFN-γ *MTB*-specific T-cell events for most subjects and time points tested, consistent with a previous study [17]. H37Rv is known to detect mycobacterial responses induced by *MTB*, BCG vaccination or exposure to environmental mycobacteria [9]. In contrast, stimulation with ESAT6/CFP10 antigens often did not result in the detection of sufficient IFN-γ *MTB*-specific cell events to allow for an accurate phenotypic analysis, and this was particularly noteworthy in HIV+ patients. It has been reported that the response to ESAT6/CFP10-specific T-cell immune response in *MTB* subjects involves only a few specific T cells, which may contribute to the low detection rates of *MTB*-specific T cells using this antigen [31]. Additionally, HIV+ patients with low CD4+ T-cell counts and percentages and an advanced disease stage [32] can have false-negative results in the interferon-gamma release assay (IGRA) [32,33,34] due to reduction in the production of *MTB*-specific IFN-γ [35]. The choice of “whole” *MTB* protein antigens, such as H37Rv or PPD, that increase the number of responders compared to ESAT6/CFP10 alone is therefore important to obtain a maximal number of valid assay results, particularly in people living with HIV. Riou et al. (2020) have also successfully tested an MTB300 peptide pool in people living with HIV with good results; furthermore, similar peptide pools with a small number of *MTB*-derived peptides have been developed [13,16,36]. The inclusion and combination of additional immunodominant *MTB* antigens into so called megapools is an approach that should therefore be considered to optimize the detection and phenotypic characterization of *MTB*-specific T cells, particularly in people living with HIV for diagnostic purposes [36,37].

Our results show that *MTB*-specific CD4+ T-cells activation and proliferation were significantly elevated in TB patients before treatment and decreased within months 2 and 6, with no significant difference between HIV+ and HIV- patients. It is noteworthy that even though the H37Rv antigen may not be specific only for *MTB*-infection [17], a TB treatment-induced decline of the activation profile on *MTB*-specific CD4+ T-cells was observed, suggesting that potential exposure to non-*MTB* mycobacteria did not interfere with the observed TB treatment-induced activation decline. Overall, these results confirm previous studies involving HIV-negative patients [10,11,12,37,38], and recent studies involving HIV/TB patients [17,19,37,39] and therefore add to the growing evidence that the TAM-TB assay can be utilized for TB diagnosis and treatment monitoring regardless of HIV infection.

The progressive depletion of systemic CD4+ T-cells is the hallmark of HIV infection [14,15,16,17,18]. ART partially reverses the loss of CD4+ T-cells and also reduces HIV-associated systemic T cell activation, as defined by single or co-expressed CD38 [16] and HLA-DR [13,17]. Consistent with Mupfumi et al. (2020), we found a significant reduction in activation marker expression within *MTB*-specific T cells [13,16,17] but not within total CD4+ T-cells at 2 months of TB treatment, which were more pronounced in the HIV coinfected patients compared to the HIV-negative TB patients. This reinforces the notion that these markers can be used to discriminate active TB and LTBI as well as to monitor declines in mycobacterial burden after two months of ART [16,17]. However, the lack of stratified analysis by ART status at baseline limited the extrapolation of our findings to HIV/TB ART naïve patients. Interestingly, although at baseline our study had the frequencies of *MTB*-specific CD4+ T-cells expressing CD38, HLA-DR and Ki67 in HIV/TB coinfected patients similar to other studies [16,17], we observed that our cohort had lower frequencies of *MTB*-specific CD4+ T-cells expressing HLA-DR compared to the study by Riou et al. (2020) [13]. Differences in the gating strategy, equipment, cellular staining procedure and type of specimen used for antigen stimulation can justify the results. Moreover, our study had 70.4% of HIV patients previously exposed to ART, of whom 30% had CD4+ T-cell counts below 100 cells/mm^3^. Riou et al. (2020) had only 38.9% of patients living with HIV on ART [13]. The HIV ART reduces the level of HIV-specific CD4+ T-cells responses [40], and this may be extrapolated to *MTB*-specific responses.

The dynamics of *MTB*-specific CD4+ T-cells activation over 6 months of TB treatment in ATB subjects have not yet been studied in relation to lung function impairment. We observed that regardless of lung impairment severity, there was a significant reduction in activation markers on *MTB*-specific T cells, suggesting that the dynamic changes within these markers are primarily linked to TB treatment-induced reductions in *MTB* bacterial load and not with lung function outcomes. We have previously shown that the slope of *MTB*-specific CD4+ T-cells activation reductions during early treatment inversely correlates with the time to *MTB* culture negativity [11]. Subjects with severe lung impairment had higher frequencies of activated *MTB*-specific CD4+ T-cells compared to those with mild lung impairment, but the difference was not statistically significant. Ravimohan et al. (2018) described that *MTB*-specific T cells secreting IFN-γ perhaps trigger the activation of effectors that result in excessive inflammation and subsequent lung disability [41]. Nevertheless, we did not measure these activation effectors in supernatants after stimulation with *MTB* antigens.

The frequency of *MTB*-specific T cells expressing CD27 did not alter over 6 months of TB treatment in the studied subjects, consistent with our previous observations [11]. CD27 expression on *MTB*-specific T cells was also independent of lung impairment. CD27 has been assessed as a surrogate of TB treatment outcome [11,14,19,42,43] and disease extent [13]. Differently to our study, Riou et al. (2020) observed that 4 weeks after TB treatment, the frequencies of *Mtb*300-specific IFN-γ CD4+ T-cells expressing CD27 increased irrespective of HIV status [13]. In the HIV-negative subjects, these frequencies are similar to those of the latent TB subjects [13]. Hence, the dynamics of CD27 expression on *MTB*-specific T cells upon treatment initiation may be influenced by the nature of the stimulation antigen used for in vitro restimulation. One study found that, at baseline, CD27 expression was strongly correlated with TB disease severity, which includes the presence of pulmonary cavities, showing a co-dependent association between these two factors [13]. Other studies found that, 12 months after treatment initiation, CD27 expression had increased in most subjects [42,44], suggesting slower dynamics of CD27 re-expression as compared to the downregulation of activation marker expression on *MTB*-specific T cells upon treatment initiation [44]. Nikitina et al. (2012) found that the accumulation of CD27 negative *MTB*-specific CD4+ T-cells in the blood is associated with lung destruction [45]. Differences in the cohort characteristics, the criteria to define lung severity, the ICS methodologies and the small sample sizes may explain the differences in the results between these studies and ours. Additionally, our study analyzed CD27-positive *MTB*-specific CD4+ T-cells, which can also drive this difference.

The limitations of this study were the relatively small sample size of subjects that were HIV/TB coinfected, ART naïve and lung function-impaired. Further, no data on the ART response, such as virus suppression or CD4+ T-cell counts after ART initiation, were available for this study. Our study did not classify *MTB*-specific CD4+ T-cells based on TNF-α production. A previous study suggested that the identification of *MTB*-specific CD4^+^ T cells defined by the production of TNF-α could be more robust than that of IFN-g and IL-2 [46]. However, TNF-α often has an unspecific background in unstimulated CD4+ T-cells during intracellular cytokine staining experiments, which complicates the differentiation of activation marker expression on truly *MTB*-specific CD4+ T-cells versus none-specific “background staining”. Another study limitation was the addition of brefeldin A without a pre-stimulation of PBMCs. Kaven et al. (2012) showed that pre-stimulation varying up to 6 h, before the addition of Golgi protein inhibitors, significantly increased the frequency of multifunctional responses CD4+ T-cells antigenic specific, rising from 0.07% to 1.31% [47].

Our study was the first to investigate TAM-TB assay phenotypic profiles in the context of pulmonary lung impairment. However, future studies with a higher number of TB patients with post-TB treatment lung impairment and HIV infection should therefore further shed light on whether or how HIV infection and lung impairment may be correlated with activation phenotypes of *MTB*-specific CD4+ T-cells.

## 4. Materials and Methods

### 4.1. Study Populations

Xpert MTB/RIF (Rifampicin)-positive study participants were recruited in Machava and Mavalane TB Research Centers in Mozambique as part of the TB sequel study [48]. Consenting TB patients with positive Xpert MTB/RIF (Cepheid, Sunnyvale, CA, USA) results were tested for HIV following the national algorithm and were followed from baseline until 12 months after TB therapy initiation during seven study visits (baseline, 14 days and 2, 4, 6, 9 and 12 months). All study participants were TB drug-sensitive. Forty-one had 6 months of a TB regimen, an intensive phase for two months with isoniazid, rifampicin, ethambutol and pyrazinamide and a continuous phase for four months with isoniazid and rifampicin. Three participants were relapsed TB cases when they were recruited into our study and received 8 months of TB treatment. The HIV-positive participants received an antiretroviral treatment composed of Tenofovir/Lamivudine/Efavirenz. During each study visit under TB treatment, liquid and solid TB culture (Lowenstein–Jensen) was performed. In those who were positive for HIV, CD4+ T-cell levels were determined at baseline. Whole blood samples were collected for Peripheral Blood Mononuclear Cells (PBMCs) processing and cryopreservation at baseline, month 2 and month 6 for all participants. Forty-six subjects were selected for our analyses based on the availability of PBMC samples and available data on HIV status, CD4+ T-cell counts, pulmonary function and spirometry at baseline, month 2 and month 6. Samples with poor cellular quality or viability were excluded (Figure 4).

### 4.2. Assessment of Lung Function and Damage

Pulmonary function impairment was categorized based on spirometry results and X-ray analysis. X-ray results were read according to Ralph score [49] by two independent readers. Spirometry was performed according to the American Thoracic Society/European Respiratory Society (ATS/ERS) guidelines. The values for Forced Ventilatory Capacity (FVC) and Forced Expiratory Volume in one second (FEV1) as well as the FEV1/FVC ratio were standardized for age, sex and height using Global Lung Function Initiative (GLI) reference equations. Pulmonary function impairment was classified as mild for an FVC and FEV1/FVC ratio >85% of predicted, moderate for an FVC or FEV1/FVC ratio 55–85% of predicted and severe for an FVC or FEV1/FVC ratio <55% predicted [23].

### 4.3. Assessment of MTB-Specific T-Cell Activation and Maturation

PBMCs were isolated using standard FICOLL centrifugation and LeucoSep tubes (Greiner Bio-One, Kremsmünster, Austria). After the last wash, the PBMCs were resuspended at a concentration of 10 million PBMC/mL/vial using inactivated Fetal Calf Serum (iFCS) supplemented with 10% Dimethylsulfoxide (DMSO) and then controlled-rate overnight cryopreserved to −80 °C using Mr. Frosty Isopropanol containers. The samples were then transferred to liquid nitrogen the next day. PBMC samples from one patient were thawed simultaneously for all three time points and were later thawed and immediately processed simultaneously using freshly prepared thawing media prewarmed to 37 °C before usage. RPMI-1640 medium with glutamax (Gibco, Invitrogen, Göteborg, Sweden), 0.2% of benzonase endonuclease (Merk, Darmstadt, Germany), 10% of heat inactivated FCS (Sigma-Aldrich, St.Louis, DE, USA) and concentration penicillin–streptomycin (Gibco, Invitrogen, Göteborg, Sweden) were added. Intracellular cytokine staining was performed using an adapted protocol from Ahmed et al., 2018 [11]. In brief, PBMCs were stimulated overnight at 37 °C and 5% CO_2_ for 20 h with Early Secretory Antigen Target Protein 6/Culture Filtrate Protein 10 (ESAT6/CFP10, 2µg/peptide/mL, Peptides & Elephants, Hennigsdorf, Germany), Purified Protein Derivative (PPD, 10 μg/mL, Serum Staten Institute, København, Denmark), *Mycobacterium tuberculosis*, strain H37Rv, whole cell lysate (the following reagent was obtained in the BEI Resources, NIAID, NIH: *Mycobacterium tuberculosis*, Strain H37Rv, Whole Cell Lysate, NR-14822) and Staphylococcal Enterotoxin B (SEB, 1 μg, Sigma-Aldrich, St.Louis, DE, USA) as a positive control and no added peptide as a negative control. A total of 50 μL of a co-stimulation cocktail composed of anti-CD49d (1 μg/mL, L25, BD, San Diego, CA, USA), anti-CD28 (1 μg/mL, L293, BD, San Diego, CA, USA) and Brefeldin A (BFA, 5 μg/mL, Sigma-Aldrich, St. Louis, DE, USA) was also added. Cells were stained with anti-CD4 APC (clone 13B8.2, Beckman Coulter, Brea, CA, USA), anti-CD27 ECD (clone 1A4CD27, Beckman Coulter, Brea, CA, USA), anti-CD38 APC fire (clone LS198-4-3, Biolegend, San Diego, CA, USA) and anti-HLA-DR PECy5 (clone Immu-357, Biolegend, San Diego, CA, USA), followed by fixation and permeabilization using FoxP3 Perm/Fix buffer and diluent (eBioscience, San Diego, CA, USA). Intracellular staining was performed using anti-IFN−γ FITC (clone B27, BD Pharmingen, San Diego, CA, USA), anti-Ki67 BV421 (clone B56, BD Pharmingen, San Diego, CA, USA) and anti-CD3 APC-A700 (clone UCHT1, Beckman Coulter, Brea, CA, USA).

Cells were acquired on a CytoFlex Flow cytometer (Beckman Coulter, Brea, CA, USA). All samples with a response in the positive control and no response in the negative control were analyzed using FlowJo_V10 (BD, San Diego, CA, USA) in a gating strategy represented in Appendix A. A subject with a positive *MTB*-specific CD4+ T-cell response was considered when: (1) the frequency of the IFN−γ+ CD4+ T-cells after H37Rv, PPD, ESAT6/CFP10 and SEB stimulation was above ≥0.03%; (2) there was a ≥2-fold increase over the negative control and (3) there were at least 20 IFN−γ+ CD4 T cell events recorded.

### 4.4. Statistical Analysis

We classified the study groups as HIV/TB coinfected or TB monoinfected based on HIV status and mild or severe pulmonary impaired based on the spirometry test. The frequencies of *MTB*-specific CD4+ T-cells expressing activation and maturation (TAM-TB) markers for each time point were compared regarding the HIV status and the severity of lung impairment using the non-parametric Mann–Whitney and Wilcoxon matched pair tests for unmatched and matched samples, respectively. Additionally, the age and Ralph scores at baseline and month 6 were compared between the HIV/TB coinfected and TB monoinfected groups using the non-parametric Mann–Whitney test. A non-parametric Spearman correlation was used to measure the correlation within the TAM-TB markers. All data were analyzed using GraphPad Prism V5 (GraphPad Software, La Jolla, CA, USA). A *p*-value < 0.05 was considered statistically significant.

## 5. Conclusions

In summary, our study supports the concept that dynamic changes in the expression of the biomarkers CD38, HLA-DR and Ki67 on *MTB*-specific CD4+ T-cells over 6 months of TB treatment can be used to monitor TB treatment outcomes among HIV-positive patients. Differences in lung function had no dramatic effects on *MTB*-specific T cell activation.

## Figures and Tables

**Figure 1 pathogens-11-01034-f001:**
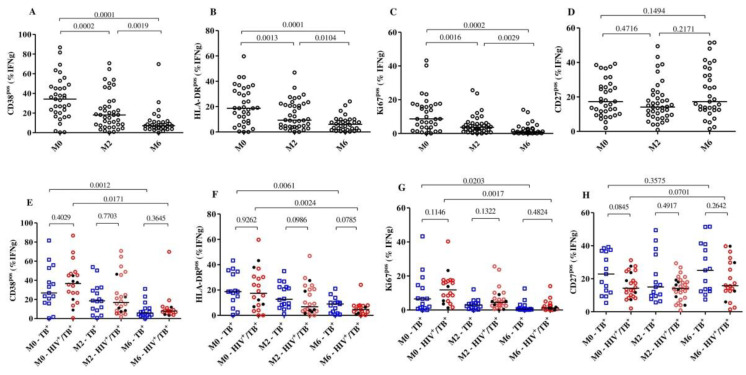
Detection of dynamic changes in the expression of the activation and maturation markers on *MTB*-specific CD4+ T-cells upon TB treatment initiation. The frequency of T cells expressing the activation markers CD38, HLA-DR, Ki67 and CD27 in all subjects (*n* = 44, (**A**–**D**)) and per group (**E**–**H**): HIV/TB coinfected (*n* = 27) and TB monoinfected (*n* = 17) at baseline (*n* = 35), 2 months (*n* = 40) and 6 (*n* = 33) months after TB treatment. The red circles and blue squares represent HIV/TB coinfected and TB monoinfected subjects, respectively. The black dots in the HIV/TB coinfected plots represent the subjects ART naïve at baseline. *MTB*-specific CD4+ T-cells were characterized after H37Rv stimulation. Bars represent medians. Statistical analyses were performed using the Mann–Whitney test for unmatched samples and with the Wilcoxon signed rank test for paired samples. *p*-values are indicated.

**Figure 2 pathogens-11-01034-f002:**
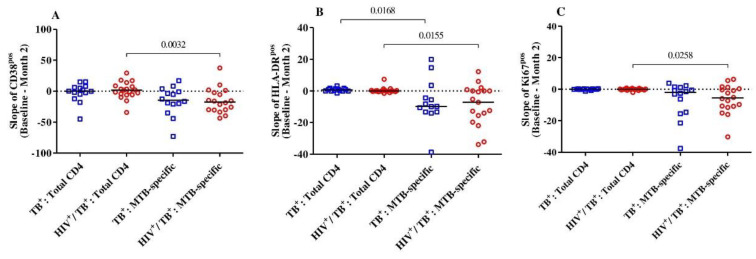
The slopes (difference from baseline to month 2) of the expression of activation markers CD38 (**A**), HLA-DR (**B**) and Ki67 (**C**) on *MTB*-specific and total CD4+ T-cells were compared between and among the HIV/TB coinfected (*n* = 19) and TB monoinfected (*n* = 14) groups. The red circles and blue squares represent HIV/TB coinfected and TB monoinfected subjects, respectively. Bars represent medians. Statistical analyses were performed using the Mann–Whitney test. *p*-values are indicated.

**Figure 3 pathogens-11-01034-f003:**
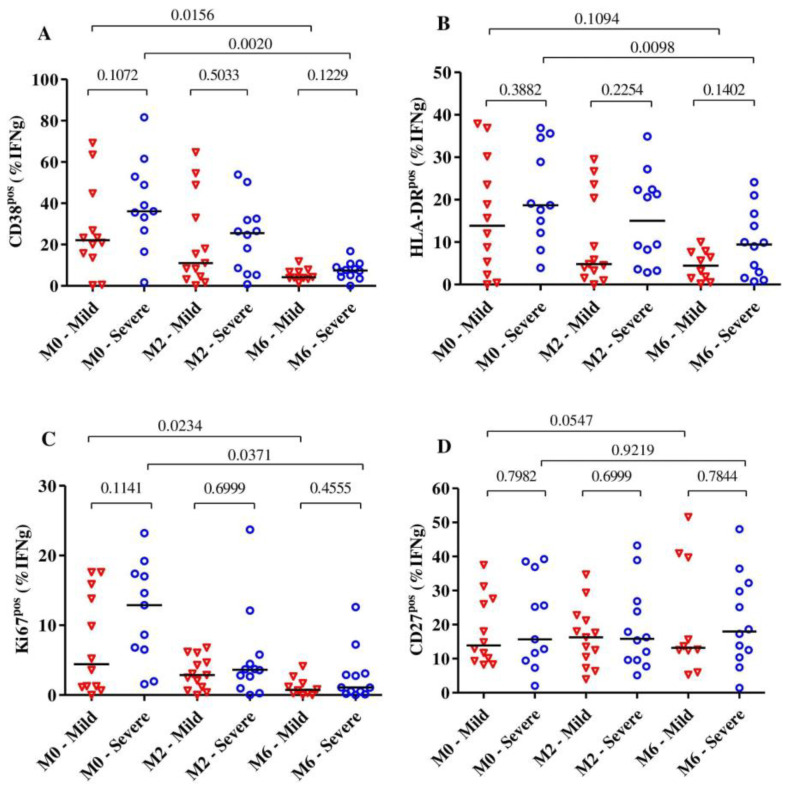
The dynamic of the expression of the activation markers CD38 (**A**), HLA-DR (**B**) and Ki67 (**C**) as well as the maturation marker CD27 (**D**) on *MTB*-specific CD4+ T-cells from the beginning to the end of TB treatment (*n* = 28). The expression of these markers was analyzed among subjects with severe (*n* = 13) and mild (*n* = 15) lung function impairment (**A**–**D**) at baseline (*n* = 23), month 2 (*n*= 25) and month 6 (*n* = 22). The blue circles and red triangles represent subjects with severe and mild lung impairment, respectively. *MTB*-specific CD4+ T-cells were characterized after H37Rv stimulation. The mild group includes subjects with mild and moderate lung impairments. Bars represent medians. Statistical analyses were performed using the Mann–Whitney test for unmatched samples and the Wilcoxon signed rank test for paired samples. *p*-values are indicated.

**Figure 4 pathogens-11-01034-f004:**
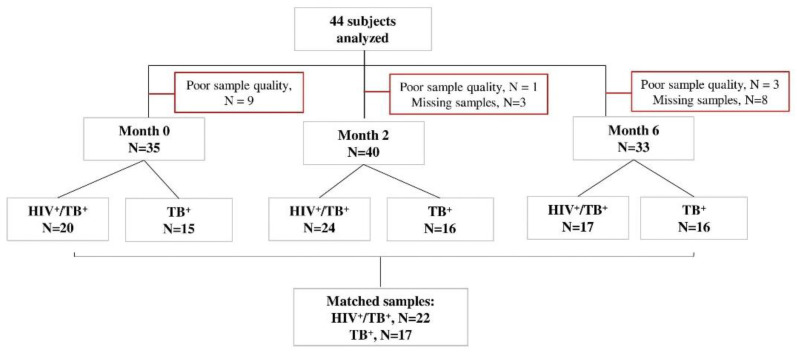
STARD flow diagram of study subjects. Peripheral blood mononuclear cell samples from TB patients with or without HIV (*n* = 44) were selected at baseline, month 2 and month 6 after TB treatment initiation, in vitro stimulated with *MTB* antigens and analyzed by flow cytometry. The number of missing visits (*n* = 11) and those excluded due to CD4 IFN-γ count <20 (poor quality samples, *n* = 13) is indicated for each time point. All samples responded to positive control antigen, SEB.

**Table 1 pathogens-11-01034-t001:** Characteristics of the study participants.

	HIV^+^/TB^+^	TB^+^	*p*-Value
N	27	17	
Median of age (range), years	37.01 (20.66–61.73)	38.57 (23.78–52.70)	0.7117
Gender (Male/Female)	19/8	12/5	
BMI ^1^ at BL (95% CI)	18.40 (17.54–20.34)	18.85 (17.94–20.35)	0.6127
Number of HIV-positive patients ART naïve	8		
Median CD4+ T-cells counts at BL (Min-Max), cells/mm^3^	279 (1–812)	------ ^6^	
Median CD4+ T-cells counts at BL on HIV ART naïve (range), cells/mm^3^	156 (66–365)	------- ^6^	
Median ratio monocytes/lymphocytes at BL (95% CI)	0.47 (0.37–0.88)	0.41 (0.38–0.63)	0.9901
Median AST ^2^ level at BL (95% CI), in U/L	22.0 (21.96–37.24)	20.0 (16.81–58.52)	0.9041
Median ALT ^3^ level at BL (95% CI), in U/L	31.0 (29.13–46.50)	26.0 (24.43–45.46)	0.6041
TB culture positivity at month 2, N	3	3	
Median ralph score at BL (Min-Max)	15.0 (5–85)	15.0 (7–55)	0.9557
Median ralph score at M6 (Min-Max)	6.0 (2–60)	10.0 (3–58)	0.1038
Presence of lung cavities at M0/M6, N	8/1	4/3	
Any lung impaired spirometry at Month 0, in %	69.57% (16/23 ^4^)	81.25% (13/16 ^5^)	
Any lung impaired spirometry at Month 6, in %	65.22% (15/23 ^1^)	76.47% (13/17)	

^1^ BMI—Body Mass Index. ^2^ AST—Aspartate Aminotransferase enzyme. ^3^ ALT—Alanine Aminotransferase enzyme. ^4^ Four subjects had spirometry data missing. ^5^ One subject had spirometry data missing. ^6^ Not applicable.

## Data Availability

Not applicable.

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
