# Peer review of "Tuberculosis Treatment Response Monitoring by the Phenotypic Characterization of MTB-Specific CD4+ T-Cells in Relation to HIV Infection Status"

_pathogens, 2022, doi:10.3390/pathogens11091034_

Round 1

Reviewer 1 Report

Manuscript ID: pathogens-1894622

The manuscript entitled "Tuberculosis treatment response monitoring by phenotypic characterization of MTB-specific T cells in relation to HIV infection status" by Sitoe and colleagues reports on the potential value of assessing the activation profile of MTB-specific CD4+ T cells for TB treatment monitoring in HIV-infected and HIV-uninfected patients. 

The findings reported in this manuscript are not novel but corroborate those of previous studies in the field.

Comments:

1- Did ART-naïve HIV/TB patients at BL also initiate HIV treatment? As both TB and ART treatment will lead to T cell “deactivation”, it will be of interest to measure the expression of CD38 and HLA-DR on total CD4 and CD3+CD4- T cells (as CD8 antibody was not used in the flow panel). Moreover, CD4 reconstitution in these patients could be monitored by reporting on the CD4 percentage using flow cytometry data (i.e., % of total CD3). The “deactivation” profile in total CD4 vs Mtb-specific CD4 T cells (BL vs M2 and BL vs M6) should be compared in all studied participants. 

2- The number of data points (n) represented on several graphs does not match “n” reported in the legend for most figures. Example: Fig 1: “The frequency of T cells expressing the activation markers CD38, HLA-DR, Ki67 and CD27 in all subjects (n=44, A-D)”. but on the figure 1A-D, M0: n=35; M2: n=40 and M6: n=33. Also, those “n” do not match the number reported in Figure 4. 

For each figure, n for each sub-group should be clearly indicated on the figure. 

3- Figure 3A is redundant and does not add much to the manuscript.

4- Supp Figure 2: correlations between expression of activation markers should be assessed before and after treatment separately.

5- For the stimulation with H37Rv whole cell lysate, was the addition of brefeldin delayed? (To allow antigen processing and presentation before the disruption of intracellular transport mechanisms).

Author Response

  • Did ART-naïve HIV/TB patients at BL also initiate HIV treatment?

R: Yes, the HIV/TB ART naïve subjects started the HIV treatment two weeks after recruitment.

  • As both TB and ART treatment will lead to T cell “deactivation”, it will be of interest to measure the expression of CD38 and HLA-DR on total CD4 and CD3+CD4- T cells (as CD8 antibody was not used in the flow panel). Moreover, CD4 reconstitution in these patients could be monitored by reporting on the CD4 percentage using flow cytometry data (i.e., % of total CD3). The “deactivation” profile in total CD4 vs Mtb-specific CD4 T cells (BL vs M2 and BL vs M6) should be compared in all studied participants. 

R: We agreed and, as suggested, we compared and presented, the “deactivation” profile in total CD4 vs MTB-specific CD4 T cells (BL vs M2 and BL vs M6) in all study participants and separated by HIV status. As follow in the supplementary figure S2, the deactivation after TB treatment is observed only in MTB-specific CD4+ T-cells, regardless to HIV status. Additionally, we highlighted in black symbols or dots, the HIV/TB coinfected subjects that were naïve at baseline, as follow in the Figure 1 and supplementary figure S2.

As shown in table 1, we had 8 ART naïve subjects which 5 had viable cellular response at BL, 6 at M2 and 3 at M6.

  • The number of data points (n) represented on several graphs does not match “n” reported in the legend for most figures. Example: Fig 1: “The frequency of T cells expressing the activation markers CD38, HLA-DR, Ki67 and CD27 in all subjects (n=44, A-D)”. but on the figure 1A-D, M0: n=35; M2: n=40 and M6: n=33. Also, those “n” do not match the number reported in Figure 4. 

For each figure, n for each sub-group should be clearly indicated on the figure.

R: We agreed and revised . Please see the figures 1, 3 and 4.

We have 44 subjects stratified by HIV status with at least one viable result based on criteria for definition of MTB-specific CD4+ T-cells response as described in the methods,  in any study time points. For those 44 subjects, 35 subjects met the criteria at baseline, 40 subjects at month 2 and 33 subjects at month 6.

For severity of lung impairment, from those 44 study participants with at least one cellular viable result, 28 had data spirometry test result available and categorized as mild, moderate and severe. Among those 28 subjects, we had 23 subjects at baseline, 25 subjects at month 2 and 22 subjects at month 4

  • Figure 3A is redundant and does not add much to the manuscript.

R: We agreed and removed the plots 3A-D and kept 3E-H. The results presented in the figure 3E-H are the novelty of our study.

  • Supp Figure 2: correlations between expression of activation markers should be assessed before and after treatment separately.

R: We agreed and revised as suggested. Please see the supplementary figure S4.

  • For the stimulation with H37Rv whole cell lysate, was the addition of brefeldin delayed? (To allow antigen processing and presentation before the disruption of intracellular transport mechanisms).

R: Following our ICS protocol, adapted from Ahmed et al (2018), we added brefeldin A at same step with H37Rv to avoid any bias, because we also did the same for PPD stimulation. Ahmed et al (2018), using PPD protein for PBMCs stimulation in similar protocol also found good results.

Reviewer 2 Report

In this manuscript authors investigated changes of activation and maturation markers on MTB-specific CD4+ T cells after tuberculosis treatment, with focus on HIV status and severity of lung impairment. Though authors observed a small population but used robust statistical tools to conclude that HIV status and lung impairment has no significant Mtb specific T cell profile. One of the major flaws of this study is missing healthy/ control population without TB or HIV. In absence of controls these observations lack reliability. For example, CD4 T cell exhaustion which is a hallmark of tuberculosis which need to be assessed to cancel out the ambiguity.

Authors should present the gating strategy for flow cytometry.

Author Response

R: We appreciate your comment that enhance our study. However, regarding the lack of health controls we partially agreed, because our primary study question was related to differences in HIV+ and HIV- active TB patients before and after initiation of TB treatment to better understand whether HIV+ patients may be affected by more continuous MTB-specific T cell activation (and hence probably not completely resolved disease) at the end of treatment.

Additionally, as suggested, we present the gating strategy for flow cytometry is presented in Supplementary Figure S5.

Round 2

Reviewer 1 Report

1-  Please, check the P-value in figure S2C between M0 and M2 (p=0.0079) and M0 and M6 (p=0.0171) for Mtb-specific CD4 T cells. It is surprising that statistical difference is stronger between M0 vs M2 compared to M0 vs M6. 

There is no mention of the activation profile of total CD4 T cells in the text (additional data added in figure S2). 

Also, in Figure S2, the legend of the x-axis needs to be corrected (i.e., (% IFNg) need to be replace by (%) as total CD4 T cells are represented on the same graphs.

2-  On supp figure S1, please check % reported on the dot plot Month 2, H37Rv, HLA-DR vs IFNg: 0.017 (HLA-DR+) vs 0.023 (HLA-DR-). Based on the number of IFN+HLADR+ events, it is possible that the frequency reported is erroneous. 

3-  Despite the ability to detect H37Rv-specific CD4 T cells, when Brefeldin is added at the onset of PBMC stimulation, such experimental design is sub-optimal (PMID: 22719990, fig 3). This need to be acknowledged as a limitation in the discussion. 

Author Response

1- Please, check the P-value in figure S2C between M0 and M2 (p=0.0079) and M0 and M6 (p=0.0171) for Mtb-specific CD4 T cells. It is surprising that statistical difference is stronger between M0 vs M2 compared to M0 vs M6. 

R: We agreed that the p-value M0 vs M6 is higher than M0 vs M2, however it was due to the outlier that was 69.8% in a median of 7.45%. With outlier, the p value was 0.0171 and excluding it, reduced to 0.0005. However, as the outlier, also an important finding, we kept this data.

There is no mention of the activation profile of total CD4 T cells in the text (additional data added in figure S2). 

R: We agreed and added as suggested. Please see page 5.

Also, in Figure S2, the legend of the x-axis needs to be corrected (i.e., (% IFNg) need to be replace by (%) as total CD4 T cells are represented on the same graphs.

R: We agreed and corrected as suggested.

2- On supp figure S1, please check % reported on the dot plot Month 2, H37Rv, HLA-DR vs IFNg: 0.017 (HLA-DR+) vs 0.023 (HLA-DR-). Based on the number of IFN+HLADR+ events, it is possible that the frequency reported is erroneous. 

R: We agreed and corrected the error.

3- Despite the ability to detect H37Rv-specific CD4 T cells, when Brefeldin is added at the onset of PBMC stimulation, such experimental design is sub-optimal (PMID: 22719990, fig 3). This needs to be acknowledged as a limitation in the discussion. 

R: We agreed and added as suggested.

Reviewer 2 Report

Authors have addressed the concerns, I endorse the manuscript.

Author Response

We revised the english language in the entire document.